# Impact of the FilmArray Rapid Multiplex PCR Assay on Clinical Outcomes of Patients with Bacteremia

**DOI:** 10.3390/diagnostics13111935

**Published:** 2023-06-01

**Authors:** Mai Okamoto, Makoto Maejima, Taichiro Goto, Takahiro Mikawa, Kazuhiro Hosaka, Yuki Nagakubo, Yosuke Hirotsu, Kenji Amemiya, Hitomi Sueki, Masao Omata

**Affiliations:** 1Department of Internal Medicine, Yamanashi Central Hospital, Kofu 400-8506, Japan; 2Clinical Laboratory Center, Yamanashi Central Hospital, Kofu 400-8506, Japan; 3Lung Cancer and Respiratory Disease Center, Yamanashi Central Hospital, Kofu 400-8506, Japan; 4Genome Analysis Center, Yamanashi Central Hospital, Kofu 400-8506, Japanmomata-tky@umin.ac.jp (M.O.); 5Department of Gastroenterology, The University of Tokyo, Tokyo 113-8655, Japan

**Keywords:** sepsis, FilmArray, blood culture, MRSA, survival

## Abstract

Bacteremia is a serious disease with a reported mortality of 30%. Appropriate antibiotic use with a prompt blood culture can improve patient survival. However, when bacterial identification tests based on conventional biochemical properties are used, it takes 2 to 3 days from positive blood culture conversion to reporting the results, which makes early intervention difficult. Recently, FilmArray (FA) multiplex PCR panel for blood culture identification was introduced to the clinical setting. In this study, we investigated the clinical impact of the FA system on decision making for treating septic diseases and its association with patients’ survival. Our hospital introduced the FA multiplex PCR panel in July 2018. In this study, blood-culture-positive cases submitted between January and October 2018 were unbiasedly included, and clinical outcomes before and after the introduction of FA were compared. The outcomes included (i) the duration of use of broad-spectrum antibiotics, (ii) the time until the start of anti-MRSA therapy to MRSA bacteremia, and (iii) sixty-day overall survival. In addition, multivariate analysis was used to identify prognostic factors. In the FA group, overall, 122 (87.8%) microorganisms were concordantly retrieved with the FA identification panel. The duration of ABPC/SBT use and the start-up time of anti-MRSA therapy to MRSA bacteremia were significantly shorter in the FA group. Sixty-day overall survival was significantly improved by utilizing FA compared with the control group. In addition, multivariate analysis identified Pitt score, Charlson score, and utilization of FA as prognostic factors. In conclusion, FA can lead to the prompt bacterial identification of bacteremia and its effective treatment, thus significantly improving survival in patients with bacteremia.

## 1. Introduction

The culture methods developed by our forerunners have led to the discovery of new microorganisms and the relationship between these microorganisms and infectious diseases. Culture-based tests have built a solid foundation for modern microbiological tests and have established a modern flow of medical care for infectious diseases that utilizes this information. While vast amounts of knowledge have been accumulated in microbiology and the treatment of infectious diseases, systemic infections such as sepsis remain difficult to treat [1,2]. Currently, most of the hospitals in Japan still use conventional biochemical properties to identify bacteria (MALDI-TOF/MS analysis). However, with such tests, it takes 2 to 3 days from positive blood culture conversion to reporting the results, making early intervention difficult [3,4].

FilmArray (FA; bioMérieux, Lyon, France) is a multiplex PCR-based desktop microbial detection system with several identification panels specifically for various pathogens including bacteria, yeasts, and viruses. By means of the FA system, microbial analysis of various specimens such as blood, sputum, or urine is completed within 1 h in a simple procedure [5,6,7,8,9,10]. The blood culture identification (BCID) panel, one of the adaptable panels for FA, diagnoses sepsis and/or systemic infections by detecting 14 bacterial species, 4 bacterial genera, 1 bacterial family, 5 yeast species, and 3 antimicrobial resistance genes (mecA, KPC, and vanA/B) in positive blood cultures [7]. The FA system has already been used frequently in Europe and the United States [11], but in Japan, our hospital was the first to introduce it in July 2018. Rapid and accurate detection of pathogenic microorganisms is supposed to be important for the treatment of sepsis and/or systemic infections [12]; however, the impact of the FA system operation on clinical outcomes and its effectiveness in real clinical practice are not fully verified.

In this study, we evaluated the concordance of the FA-BCID panel with MALDI-TOF/MS analysis for detecting pathogenic microorganisms in positive blood culture samples and assessed its clinical utility. Furthermore, in order to verify the clinical benefit of treatment for patients with bacteremia, we conducted a comparative study of clinical outcomes before and after the introduction of this device at our hospital.

## 2. Methods

### 2.1. Patients

This single-center, pre-postintervention study was performed at Yamanashi Central Hospital in Yamanashi, Japan. All study participants provided informed consent, and the study was approved by the Institutional Review Board at our institution (protocol code Clin18-5). All adult patients (17 years or older) with a positive blood culture(s) between 1 January and 31 October of the year 2018 were included in the study unless they met any exclusion criteria. Patients who had a history of a previously positive blood culture of the same organism, those whose blood cultures were deemed to contain contaminants, and those who were not admitted to our hospital were excluded from the study (Figure 1).

FA BCID technology was introduced at our hospital on 1 July 2018. Patients with bloodstream infections (BSI) prior to the implementation of the blood culture identification panel (BCID) technology (control group: *n* = 156) were compared to patients with BSI after implementation of BCID (FA group: *n* = 139). Only the first positive culture for each patient was included during the study period; any subsequent episode of BSI was excluded.

### 2.2. Laboratory Testing

For all patients, the local standard-of-care for identification and AST of bacteria from positive blood cultures were performed, including Gram stain, standard subculture, species identification by matrix-assisted laser desorption/ionization mass spectrometry (MALDI-TOF MS), and AST using broth microdilution or agar dilution. Gram stain results were immediately communicated by phone while identification and AST results were transferred upon availability into the patients’ computerized medical records. During the postintervention period, testing was also performed using the FilmArray system on the first positive blood culture bottle of each episode within an hour following the alarm signal for bacterial growth detection, and results were reported in the electronic medical record.

### 2.3. Antimicrobial Stewardship

All patients in both groups underwent prospective audit and feedback by institutional antimicrobial stewardship (AS) programs. In both arms, the AS physician or pharmacist was notified by page at the time of every positive blood culture, organism identification, and AST, regardless of testing method. The AS provider reviewed the record and contacted the primary care service by telephone if adjustment of therapy was required. Timing and type of AS recommendations were at the discretion of the AS clinicians. Acceptance of AS recommendations was at the discretion of treatment providers.

### 2.4. Data Handling

Microbiological data of the positive blood culture episodes were recorded from the laboratory information system and patients’ medical records were reviewed for collection of demographic and clinical characteristics, blood culture pathogens, and antibiotic treatment data as well as their follow-up data. Differentiation of positive blood culture episodes into contamination or BSI was defined according to the US Centers for Disease Control and Prevention/National Healthcare Safety Network definitions of bloodstream infection events [13].

### 2.5. Outcomes

The evaluated clinical outcome was the median duration of empirical therapy and their prognosis after the treatment. The empirical treatment was defined as the antimicrobial drug administered before availability of any laboratory results. Treatment modifications were labeled as a de-escalation of the empirical treatment. In accordance with the ICU restrictive antimicrobial policy, an empirical antibiotic was, as a rule, given to patients with suspected sepsis. Depending on the recent medical history of the patient, the suspected infectious source, and previous antimicrobial therapy, the treatment usually began with empirical therapy. All data on BSI and antibiotic management were ultimately reviewed by an adjudication committee composed of an intensive care practitioner and a microbiologist. This study aimed to demonstrate a reduced time of empirical therapy in the FA group compared with the control group. Patients with septic infections caused by MRSA may also benefit from rapid AST. FilmArray allows for rapid detection of the mecA gene, which is associated with drug resistance in MRSA, and the clinical significance of this was confirmed in this study. In parallel, identification and resistance detection performances of the FA-BCID panel were evaluated.

On the other hand, we compared the prognosis between the control and the FA groups. In addition, we attempted to identify factors that correlated with prognosis among age, gender, Gram-positive cocci/-negative bacilli, Charlson score, Pitt score, and bacterial identification methods by multivariate analysis. All time measurements started at the time when the blood culture was determined to be positive.

### 2.6. Statistical Analyses

Microbiological data of the positive blood culture episodes were recorded from the laboratory information system and patients’ medical records were reviewed for collection of demographic and clinical characteristics as well as antibiotic treatment data. Classification of positive blood culture episodes into BSI or contamination was defined according to the US Centers for Disease Control and Prevention/National Healthcare Safety Network definitions of bloodstream infection events. Continuous variables are presented as the mean ± SD and compared using unpaired Student’s *t* tests. Chi-square tests were used to compare the categorical data between the groups. Overall survival time was defined as the period from the day of diagnosis of sepsis to the day of death or that of the last follow-up evaluation. Survival was assessed using the Kaplan–Meier method, and comparisons among the survival curves were made using the log-rank test. To determine the predictors of survival within the cohort, we constructed Cox proportional hazards models including each variable of interest. Multivariate analyses were performed using the JMP function in the SAS software (SAS Institute, Cary, NC, USA). The *p* values less than 0.05 on two-tailed analyses were considered to denote statistical significance.

## 3. Results

### 3.1. Patient Characteristics

The study inclusion process of the critically ill patients with a positive blood culture episode is presented in Figure 1.

Of the 533 patients with positive blood culture, 262 patients between 1 January 2018 and 30 June 2018 and 271 patients between 1 July 2018 and 31 October 2018 were grouped, respectively. Of these, 156 patients in the control group and 139 patients in the FA group, excluding patients who met the exclusion criteria, were compared for clinical outcomes.

Comparing the characteristics of the patients, there were no significant differences between the two groups in terms of gender, Charlson score, Pitt bacteremia score, empiric vancomycin use, and nursing home admission, but the patients in the control group were significantly older. Regarding underlying diseases, collagen, and liver diseases, immunosuppressant use was significantly more common in the FA group (Table 1).

### 3.2. FA-BCID Microbiological Performances

Microorganisms were detected in all cases in the control group (156 cases) and the FA group (139 cases) by the conventional organism identification. Results of detailed MALDI-TOF/MS identification and corresponding FA-BCID are presented in Table 2. Overall, 122 (87.8%) microorganisms were concordantly retrieved with the FA-BCID in 139 patients. A total of 17 (14.8%) strains gave a “no organism detected” FA-BCID result, all of which are off-panel strains, as follows; 6 *Bacillus *spp., 2 *Bacteroides fragilis*, 2 *Corynebacterium striatum*, 2 *Vibrio vulnificus*, 2 *Yersinia Enterocolitica*, 1 *Campylobacter *sp., 1 *Citrobacter freundii*, and 1 *Clostridium perfringens*. The sensitivity of the FA-BCID test in accordance with its on-panel microorganisms was 100.0% (122/122). The molecular test detected the mecA gene in 14 MRSA and 2 MRCNS coagulase negative staphylococci with both a sensitivity and specificity of 100%.

### 3.3. Treatment Selection According to the Analyses

Treatment with broad-spectrum antibiotics was started in 44 patients in the control group and 38 patients in the FA group followed by switching antibiotic and de-escalation after bacterial identification. Figure 2 shows a comparison of the usage time of broad-spectrum antibiotics in these cases. The duration of ABPC/SBT (ampicillin/sulbactam) use was significantly shorter in the FA group than in the control group. There was no significant difference between the two groups in PIPC/TAZ (piperacillin/tazobactam), CFPM (cefepime) and MEPM (meropenem).

Furthermore, when comparing the time until the start of vancomycin administration for MRSA sepsis between the two groups, it was significantly shorter in the FA group (Figure 3).

### 3.4. Clinical Outcomes

In the FilmArray group, bacterial species identification results were obtained within 2–3 h after blood culture positivity, while in the control group, these results were obtained 1–3 days later. The number of patients who died during the waiting period for identification results was 0 in the FilmArray group, whereas it was 7 in the control group.

Survival after the onset of sepsis was compared between the FA and the control groups (Figure 4). First, the prognosis within 60 days from the diagnosis of sepsis was significantly better in the FA group.

Next, a multivariate analysis was performed on the factors involved in the prognosis among age, gender, Charlson score, Pitt score, bacteremia species identification method (before and after FA introduction), and bacteremia-causing bacteria (GPC vs. GNR). As a result, three factors, Charlson score, Pitt score, and bacteremia species identification method, were identified as prognostic factors (Table 3).

## 4. Discussion

The results of this study showed a comparison of treatment strategies and clinical outcomes in sepsis patients before and after the introduction of FilmArray in our hospital. It was demonstrated that, in patients after the introduction of FilmArray, the time to initiate appropriate antibiotic therapy was shortened, and the prognosis was improved.

Life-threatening BSI require prompt and appropriate diagnosis [14]. However, blood cultures, which are essential for diagnosis, take 1 to several days to become positive, and microbial identification and drug susceptibility testing require about 2 to 3 more days [15,16]. The introduction of genetic testing to the diagnosis of BSI is also attracting attention as an effective means that can significantly change the treatment policy [17,18,19]. When multiple strains of bacteria are present in a blood culture bottle, the growth of bacteria may vary, and slow-growing bacteria may not be detected [20]. However, in our analyses, there was no discrepancy between the on-panel strains and the blood culture results, indicating that the diagnostic accuracy of the FA system was very high.

Clinically, empirical therapy is performed considering the patient’s clinical symptoms, blood test/imaging findings, and epidemiological information until the results of microbiological tests are reported [21]. During empirical therapy, inappropriate treatment is given in a certain percentage, so the quality of diagnosis and treatment of individual cases will improve through the introduction of genetic testing that speeds up the process. In this study, the administration duration of ABPC/SBT was significantly shortened in the FilmArray group, and there was a trend towards the overall shortening of empiric therapy, although the sample size was small and did not reach statistical significance. Furthermore, in the FilmArray group, the rapid detection of the drug-resistant gene mecA led to a significantly earlier initiation of vancomycin administration. This indicates that the use of FilmArray facilitated an earlier switch from ineffective antibiotics.

In addition, this study is the first to examine the effects of FA, a bacterial identification method for bacteremia, on prognosis. Importantly, the genetic bacterial identification method has been shown to significantly improve the prognosis of sepsis patients. In other words, FA not only contributes to the rapid identification of bacteria, but also enables a rapid and appropriate treatment system, and as a result, the ultimate clinical outcome of improving patient prognosis was also achieved.

The systematic review conducted by Anton-Vazquez et al. included six randomized trials involving a total of 1638 participants [22]. For rapid antimicrobial susceptibility testing compared to conventional methods, there was little or no difference in mortality between the groups (RR 1.10, 95% CI 0.82 to 1.46; 6 RCTs, 1638 participants; low-certainty evidence). Meanwhile, their confidence in the results was limited for the following reasons: (1) the reported numbers of deaths in the studies were insufficient to establish a significant difference; (2) there was considerable variation in the tests used and the results obtained from the studies; and (3) the studies lacked an adequate number of participants to draw firm conclusions. Anton-Vazquez et al. ultimately predicted that further research is likely to alter these results, as stated in their description [22]. Upon reviewing the six randomized trials included in this review, it is evident that, with the exception of one study conducted at two centers [23], all others were single institution studies [24,25,26,27,28]. The sample sizes for both the intervention arm (*n* = 118.3 ± 75.0) and the control arm (*n* = 119.3 ± 78.7) were relatively small, with the number of deaths in the intervention arm (*n* = 13.3 ± 8.9) and control arm (*n* = 11.5 ± 7.3) also being low [23,24,25,26,27,28]. Furthermore, due to the nature of the study intervention, blinding was not implemented in any of the RCTs, indicating that the study design may have included a certain degree of bias. Interestingly, another systematic review conducted by Timbrook et al. concluded that rapid diagnostic testing, which included identification of the organism and detection of resistance mechanisms, or both, was associated with reduced time to effective antibiotics, shorter length of stay, and decreased mortality rates when implemented alongside an antimicrobial stewardship program [29]. However, in this review, the majority of studies were observational studies conducted in a pre- and post-intervention design, similar to our own study [30,31,32,33,34,35,36,37,38,39]. It is worth considering the discrepancy between the findings of many pre-post studies, which showed improvement in mortality with the introduction of molecular rapid diagnostic testing, and the lack of mortality improvement in the aforementioned six randomized trials. In actual clinical practice, patients with sepsis are often critically ill, and it is unlikely that all patients would have been enrolled in these RCTs from an ethical standpoint. There is a possibility that the most severely ill patient population may not have been included in these RCTs, leading to potential selection bias. It is speculated that such selection bias may have influenced the results of the RCT trials.

There are several important treatment factors that determine the mortality of sepsis, including the search for the infectious focus, control of the infection source (such as treating the underlying disease or drainage), treatment of complicating conditions such as septic shock or systemic inflammatory response syndrome (SIRS), and treatment of concurrent diseases such as diabetes or cancer. Even if the time to de-escalation can be shortened, we believe that solving all the problems associated with sepsis cannot be achieved in this way. On the other hand, while other treatments can be addressed immediately in parallel with the patient’s condition, the issue with appropriate antibiotic therapy is that there may be delays in the treatment of several days due to the time required for lab works for bacterial identification, and we consider this to be a problem. In other words, the time required for bacterial identification can become a bottleneck in establishing the optimal combination of antimicrobial therapy in harmony with other treatments for sepsis. In this study, we investigated whether minimizing the timing lag between optimal antimicrobial therapy and other concurrent therapies would achieve the ultimate goal of sepsis therapy: improvement in mortality. The results of this study showed a general trend of shortened time to antibiotic de-escalation, and significant reduction in the time to initiate vancomycin treatment for MRSA, indicating that such improvements can contribute to improved mortality by synergistic effects with other treatments at an early stage.

As a limitation of this study, firstly, the two groups were not randomized, and the research design included a high degree of bias. There were some differences in baseline characteristics between the cohorts, including the timing of treatment. Secondly, this study did not analyze the time to bacterial identification, length of hospital stays, or length of stays in the ICU. A detailed analysis of cost has not been conducted either. Although this is a cohort study conducted at a single institution and the sample size is modest, we believe that it will provide useful insight regarding the operation and effectiveness of FA in line with actual clinical practice.

## 5. Conclusions

In conclusion, FA can lead to prompt bacterial identification of bacteremia and effective treatment, thus significantly improving survival in patients with bacteremia.

## Figures and Tables

**Figure 1 diagnostics-13-01935-f001:**
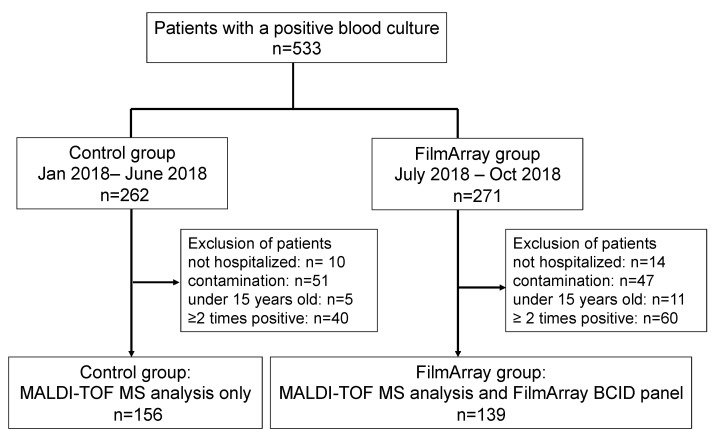
Study inclusion process of patients with a positive blood culture episode among control and FA groups. Abbreviations: MALDI-TOF/MS, matrix-assisted laser desorption ionization time-of-flight mass spectrometry; BCID, blood culture identification.

**Figure 2 diagnostics-13-01935-f002:**
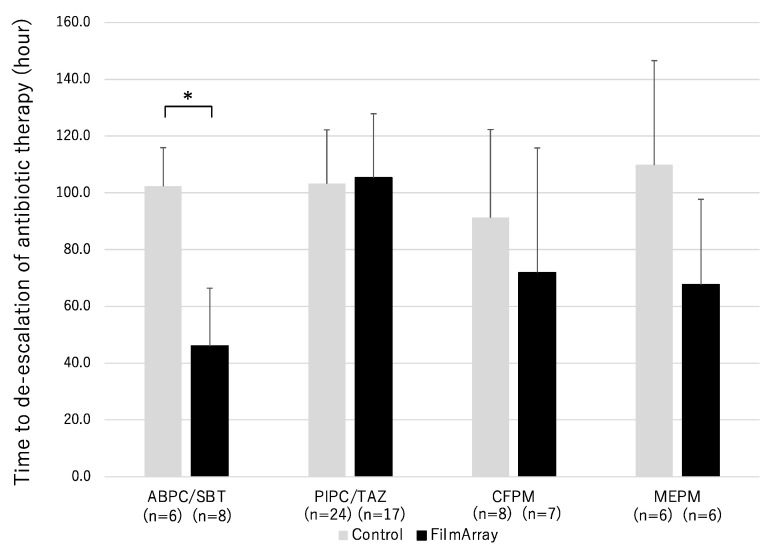
Time until de-escalation of antibiotic therapy. Gray bar, control group. Black bar, FA group. * *p* < 0.05. ABPC/SBT: ampicillin/sulbactam, PIPC/TAZ: piperacillin/tazobactam, CFPM: cefepime, MEPM: meropenem.

**Figure 3 diagnostics-13-01935-f003:**
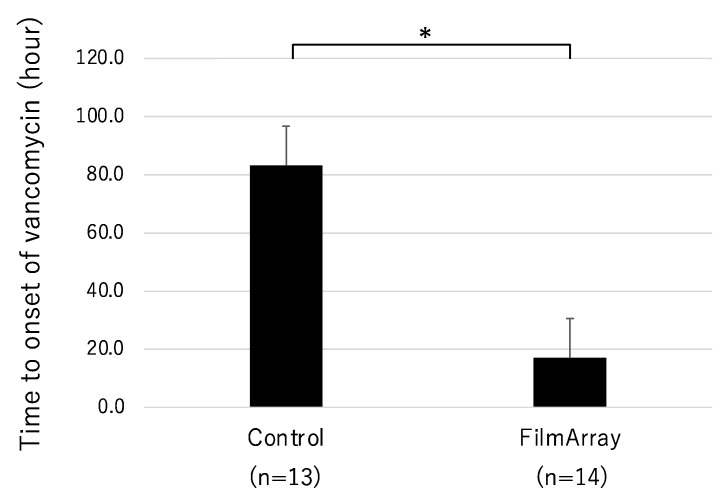
Time until onset of vancomycin in MRSA sepsis. * *p* < 0.05.

**Figure 4 diagnostics-13-01935-f004:**
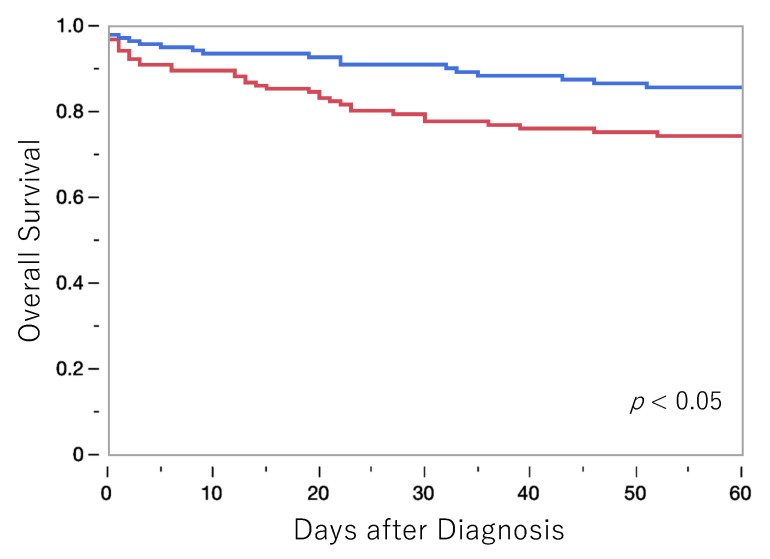
Comparison of short-term prognosis of 60 days after sepsis diagnosis. The prognosis was significantly better in the FA group. Blue line, FA group; red line, control group.

**Table 1 diagnostics-13-01935-t001:** Characteristics of the patients.

	Control (*n* = 156)	FilmArray (*n* = 139)	*p* Value
Male/Female	84/72	88/51	0.10
Age	74.9 ± 11.7	70.0 ± 14.9	0.002
Nursing home	13 (8.3)	15 (10.8)	0.47
Charlson score	5.7 ± 2.5	5.4 ± 2.3	0.33
Myocardium infarction	8 (5.1)	11 (7.9)	0.33
Chronic heart failure	14 (9.0)	13 (9.4)	0.91
Chronic vascular disease	32 (20.5)	20 (14.4)	0.17
Leukemia	6 (3.8)	8 (5.8)	0.44
Collagen disease	1 (0.6)	7 (5.0)	0.01
Diabetes Melitus	33 (21.2)	37 (26.6)	0.27
Solid tumor	34 (21.8)	29 (20.9)	0.85
Liver disease	8 (5.1)	16 (11.5)	0.04
AIDS *	1 (0.6)	0 (0)	0.26
Immunosuppressant	11 (7.1)	20 (14.4)	0.04
Pitt score	1.5 ± 1.7	1.2 ± 1.5	0.3
Empiric vancomycin use	16 (10.3)	8 (5.8)	0.15

* AIDS: acquired immunodeficiency syndrome.

**Table 2 diagnostics-13-01935-t002:** Detected microorganisms in each group.

Identified Microorganisms	Control Group	FA Group	
*Bacillus* spp.	3	6	*
*Bacteroides fragilis*	1	2	*
*Bacteroides ovatus*	1		
*Campylobacter* sp.		1	*
*Candida albicans*	1		
*Capnocytophaga* sp.	1		
*Citrobacter freundii*		1	*
*Clostridium perfringens*	1	1	*
*Corynebacterium jeikeium*	1		
*Corynebacterium striatum*	1	2	*
*Eggerthella lenta*	1		
*Enterobacter aerogenes*	3	1	
*Enterobacter aerogenes AmpC*	1		
*Enterobacter cloacae*	2	4	
*Enterococcus faecalis*	3	4	
*Enterococcus faecium*	1		
*Eschelichia coli*	29	30	
*Escherichia coli ESBLs*	8	5	
*Klebsiella oxytoca*	1	2	
*Klebsiella pneumoniae*	11	14	
*Listeria monocytogenes*	1		
*Moraxella catarhalis*	1		
*Prevotella melaninogenica*	1	1	
*Proteus mirabilis*	3	3	
*Pseudomonas aeruginosa*	3	2	
*Raoultella planticola*	1		
*Serratia marcescens*	1	1	
*Staphylococcus aureus MRSA*	13	14	
*Staphylococcus aureus MSSA*	26	21	
*Staphylococcus capitis*	1		
*Staphylococcus epidermidis*	8	8	
*Staphylococcus haemolyticus*	2		
*Staphylococcus hominis*	1		
*Staphylococcus hominis MRCNS*	1		
*Staphylococcus lugdunensis*	1		
*Staphylococcus warneri*	1		
*Streptococcus agalactiae*	6	5	
*Streptococcus constellatus*	2		
*Streptococcus dysgalactiae sub. equisimilis*	3	3	
*Streptococcus intermedius*	1		
*Streptococcus mitis*	1	1	
*Streptococcus oralis*	1	2	
*Streptococcus parasanguinis*	1		
*Streptococcus pneumoniae*	4	1	
*Streptococcus pyogenes*	2		
*Vibrio vulnificus*		2	*
*Yersinia enterocolitica*		2	*
total number	156	139	

* Microorganisms detected only by MALDI-TOF/MS in the FA group.

**Table 3 diagnostics-13-01935-t003:** Multivariate analyses for overall survival in patients with sepsis.

Variables	*n*	Survival	
		Hazard Ratio (95%CI)	*p* Value
Sex			
Male	174	1.30 (0.85–1.99)	0.22
Female	121	1	
Age			
17–69	90	1.33 (0.77–2.32)	0.31
70–79	101	0.90 (0.54–1.48)	0.66
80–96	104	1	
Gram-stain			
Gram positive cocci	139	0.97 (0.63–1.48)	0.87
Gram negative rods	132	1	
Charlson score			
0–3	48	0.39 (0.19–0.81)	0.01
4–6	165	0.60 (0.39–0.92)	0.02
7–17	82	1	
Pitt score			
0	115	0.47 (0.29–0.77)	0.003
1–2	120	0.31 (0.18–0.53)	<0.0001
3–9	60	1	
Bacterial Identification			
FA + MALDI-TOF/MS	139	0.65 (0.43–0.98)	0.04
MALDI-TOF/MS only	156	1	

CI: confidence interval, MALDI-TOF/MS: matrix-assisted laser desorption ionization time-of-flight mass spectrometry.

## Data Availability

The data presented in this study are available on request from the corresponding author. The data are not publicly available due to them containing information that could compromise research participant privacy/consent.

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
