# Peer review of "Impact of the FilmArray Rapid Multiplex PCR Assay on Clinical Outcomes of Patients with Bacteremia"

_diagnostics, 2023, doi:10.3390/diagnostics13111935_

Round 1

Reviewer 1 Report (Previous Reviewer 2)

In this revised manuscript, the authors made significant revisions based on feedback from myself and another reviewer. While I appreciate their revisions, I still have a couple of comments.

1.       Regarding my previous comment 4 about why they reported time to vancomycin initiation while they compared duration of other broad-spectrum antibiotics. I suspect the authors’ institution anti-MRSA drugs are not used as an empiric therapy routinely. FA panel can be useful in 2 ways – early initiation if vancomycin was not empirically started but FA panel suggested MRSA, and early discontinuation of vancomycin when it was started empirically and there is no evidence of MRSA infection. In the US where prevalence of MRSA is high, vancomycin is commonly used as a part of empiric therapy. As a physician in such an environment, I could not easily understand why they compared time to vancomycin initiation rather than duration of them.

To make that point clearer, I would recommend to add % of patients who were started vancomycin empirically in table 1. In addition, the added paragraph in introduction (Lines 50-59) seem redundant and the sentence “Therefore, it is desirable to use anti-MRSA drugs only after MRSA has been confirmed as the causative pathogen” can be misleading. I would suggest removing the paragraph.

2.       About my previous comment 5 about statistical analysis. The authors tried to use Weibull distribution as the suitable distribution. I highly doubt they can use Weibull distribution in their study. Weibull distribution is the parametric approach which assumes monotonic hazard increase or decrease over time. In their study, we can biologically assume mortality hazard would be very high during acute illness but after acute illness mortality hazard would significantly decrease and probably slowly increase over 3 years due to aging. In that case, they cannot assume monotonic hazard change. I do believe two possible statistical methods they can use to accurately analyze the longitudinal analysis would be to use time-varying covariate, or separate the analysis to two period (acute versus post-acute).

That being said, I do not believe it is biologically plausible to think FA panel affect the long-term outcome. And I do not think it very important. In my opinion, this study is already meaningful without long-term outcome. For that reason, I would recommend just removing long-term outcome analysis to make it simple rather than trying to more complicated analysis.

Author Response

Reviewer 1

  1. To make that point clearer, I would recommend to add % of patients who were started vancomycin empirically in table 1. In addition, the added paragraph in introduction (Lines 50-59) seem redundant and the sentence “Therefore, it is desirable to use anti-MRSA drugs only after MRSA has been confirmed as the causative pathogen” can be misleading. I would suggest removing the paragraph.

Response: Thank you for your thoughtful comments. According to the reviewer’s suggestion, the percentage of patients who started treatment with vancomycin empirically was added to Table 1. We also agree with the reviewer that the paragraph may be misleading, so the paragraph was deleted.

  1. That being said, I do not believe it is biologically plausible to think FA panel affect the long-term outcome. And I do not think it very important. In my opinion, this study is already meaningful without long-term outcome. For that reason, I would recommend just removing long-term outcome analysis to make it simple rather than trying to more complicated analysis.

Response: We agree with the reviewer. The description of long-term prognostic analysis was removed from the text.

Owing to your thoughtful comments, our manuscript has much improved. We really appreciate your consideration.

Reviewer 2 Report (New Reviewer)

In the manuscript titled ‘Impact of the FilmArray rapid multiplex PCR assay on clinical outcomes of patients with bacteremia’ the authors show that in patients with bacteremia FA bacterial identification was rapid which lead to precise treatment in turn resulting both short and long term survival.

The manuscript is written in a sound scientific language, experiments and data analysis performed accurately.

I have the following suggestions to the authors for the revision of the manuscript.

Line 46:…... ‘systemic infections remain highly lethal’- reframe the last part of this sentence

Line 51 and 51: …….most common causative organism- in bacteremia ? also add a reference at the end of this sentence

Line 142: …..detected positive by the incubators

Already mentioned in the previous section

Author Response

Reviewer 2

I have the following suggestions to the authors for the revision of the manuscript.

Line 46:…... ‘systemic infections remain highly lethal’- reframe the last part of this sentence

Line 51 and 51: …….most common causative organism- in bacteremia ? also add a reference at the end of this sentence

Line 142: …..detected positive by the incubators

Response: Thank you for your thoughtful comments.

According to the reviewer’s suggestion, we have rephrased the relevant sections, as follows.

“systemic infections such as sepsis remain difficult to treat.”

“All time measurements started at the time when the blood culture was determined to be positive.”

As to the reviewer’s second point, we have removed this paragraph following the suggestions of another reviewer.

Reviewer 3 Report (New Reviewer)

In this study, the authors evaluated concordance of the FA-BCID panel introduced by their hospital with MALDI-TOF/MS analysis at detecting pathogenic microorganisms in positive blood culture samples and assessed its clinical utility. To verify the clinical benefit of treatment for patients with bacteremia, they conducted a comparative study of clinical outcomes (including long-term prognosis) before and after the introduction of this device at their hospital. However, there are some doubts in that study. (1) What was the basis for the authors' grouping? (2) In the line 162-163: “Of the 533 patients with positive blood culture, 156 patients between 1 January 2018 and 30 June 2018 and 271 patients between 1 July 2018 and 31 October 2018.” The total number of patients does not seem right and the authors need to verify it. 

Minor editing of English language required

Author Response

Reviewer 3

However, there are some doubts in that study. (1) What was the basis for the authors' grouping? (2) In the line 162-163: “Of the 533 patients with positive blood culture, 156 patients between 1 January 2018 and 30 June 2018 and 271 patients between 1 July 2018 and 31 October 2018.” The total number of patients does not seem right and the authors need to verify it. 

Response:

Thank you for your comments.

I checked the manuscript and found that 156 patients was an error for 262 patients. As shown in Figure 1, 156 is the number of patients excluding those who met the exclusion criteria out of 262 patients. We corrected the number in the revised manuscript.

We really appreciate your consideration.

Round 2

Reviewer 1 Report (Previous Reviewer 2)

The authors made satisfactory revisions in response to the comments from me and other reviewers. 

This manuscript is a resubmission of an earlier submission. The following is a list of the peer review reports and author responses from that submission.

Round 1

Reviewer 1 Report

The statistical results seem to have serious problems, which is unacceptable. For example, many of the p values in Table 1 are wrong.

The data in Table 1 of the article is wrong, which belongs to scientific research integrity.

Reviewer 2 Report

This is a single-center pre-post quasi-experimental study to assess the performance of the FilmArray multiplex PCR access on blood culture and patients’ outcome. They compared 156 patients in pre-intervention period (BCx identified with MALDI-TOF MS) and 139 patients in the intervention period (BCx identified with a combination of MALDI-TOF MS and FilmArray). Their findings include good sensitivity of FilmArray, shorter duration of empiric use of ampicillin/sulbactam, and better short-term and long-term mortality of patients in intervention period.

There are studies which evaluated the performance of rapid BCx diagnostics, and there is a recent Cochrane review of RCTs about this issue (10.1002/14651858.CD013235.pub2.). The review failed to show the rapid diagnostic’s ability to improve mortality, time to discharge and time to appropriate antibiotics. This study’s findings are somewhat conflict with the Cochrane review.

The study itself seemed to be fairly conducted. It would be good to report how the implementation of FilmArray affect the clinical practice in Japan where it has not been used. There are several clarifications needed in the method section but results are nicely reported. Discussion is very poor. Currently it includes general discussion with some of them repetitions in the background section. For this type of study which reports findings from observation/trial, they should discuss the findings from their observation, how they are biologically plausible, how they are alike or different from findings from previous study, and what are possible explanations if those findings are different from previous study, etc.

My specific comments are as below,

1.       Figure 1 – in the first box “Critically ill patients with a positive blood culture” – it is somewhat different from what they wrote in method section.  Not all patients with positive blood culture are critically ill. Please clarify and correct.

2.       Methods, 2.2 outcomes – the authors used two words “broad spectrum antibiotics” and “empirical therapy”. Not sure they used them interchangeably, but I would recommend to consistently use “empirical therapy”. In the same section (Lines 96-108), did they explain their practice pattern in the hospital, or did they explain how they collected the data?

3.       Methods, 2.2 outcomes – “In accordance with the ICU restrictive antimicrobial policy, an empirical antibiotic was exclusively given to patients with suspected sepsis.” Does it mean providers are not allowed to use empirical antibiotics for stable patients but suspected infection?

4.       Methods, 2.2 outcomes – authors should include why they compared time to initiation of vancomycin while they compared duration of other empirical antibiotics.

5.       Methods, 2.3 statistical analysis – they used Cox proportional hazard model to assess the predictors of long-term mortality in 3 years. I do not think it appropriate to use this method. Cox proportional hazard model requires proportional hazards assumption which assumes the hazard is constant over time. The benefit from the use of FilmArray would not last for 3 years in a same degree. For time-to-event analysis with violated proportional hazard assumption, I would suggest to use Cox regression model with time-varying coefficients, to calculate the time-dependent hazard ratios (https://dx.doi.org/10.1016/S0895-4356(01)00363-8).

6.       How they dealt with patients who died before isolates were fully identified? Those patients were not affected by the methods of BCx identification. In my opinion, those patients should be removed from mortality analysis.

7.       Results. Lines 140-144. This part is a repetition of the method section and should be removed.

8.       Results, 3.3 Treatment selection according to the analysis – Abbreviated antibiotic names, ABPPC/SBT, PIPC/TAZ, CFPM and MEPM should be spelled out at their first appearance.

9.       Table 2, footnote – what is group 2? Please clarify.

10.   Discussion – as I stated in the general comment, currently it includes very general discussion which should be stated in background section. Some of the findings in this study, including mortality benefit, are conflicting with previous studies including Cochrane review. They should focus on their findings and discuss about them.

11.   The mortality benefit is somewhat questionable to me. They did not assess important clinical factors such as source of bacteremia, duration of bacteremia, source control etc. And the method of BCx identification should not directly affect the mortality, but rather affect the mortality indirectly via practice change of treating provider (such as use of appropriate antibiotic sooner). The discussion why they observed the mortality benefit and possible explanations should be included in the discussion/limitation section.